# Applying Social Network Analysis to Identify Project Critical Success Factors

**Marco Nunes** [1,*] and **António Abreu** [2,*]

1   Industrial Engineering Department, University of Beira Interior, 6201-001 Covilhã, Portugal
2   Mechanical Engineering Department, Polytechnic Institute of Lisbon and CTS Uninova,
    1959-007 Lisbon, Portugal
*   Correspondence: marco.nunes@tetrapak.com or D2317@ubi.pt (M.N.); ajfa@dem.isel.ipl.pt (A.A.)

**Abstract:** A key challenge in project management is to understand to which extent the dynamic interactions between the different project people—through formal and informal networks of collaboration that temporarily emerge across a project´s lifecycle—throughout all the phases of a project lifecycle, influence a project's outcome. This challenge has been a growing concern to organizations that deliver projects, due their huge impact in economic, environmental, and social sustainability. In this work, a heuristic two-part model, supported with three scientific fields—project management, risk management, and social network analysis—is proposed, to uncover and measure the extent to which the dynamic interactions of project people—as they work through networks of collaboration—across all the phases of a project lifecycle, influence a project's outcome, by first identifying critical success factors regarding five general project collaboration types ((1) communication and insight, (2) internal and cross collaboration, (3) know-how and power sharing, (4) clustering, and (5) teamwork efficiency) by analyzing delivered projects, and second, using those identified critical success factors to provide guidance in upcoming projects regarding the five project collaboration types.

**Keywords:** project management; risk management; social network analysis; project outcome likelihood; project lifecycle; project critical success factors; social capital; people data management; competitive advantage; sustainability

## 1. Introduction

If organizations want to achieve a sustainable competitive advantage, in today´s turbulent and instable business environment, they need to craft strategies that allow them to boost performance and innovation [1,2]. Both are strongly influenced by how an organization´s core apex guides and motivates the whole organizational structure in order to overcome some major constraints, such as geographic barriers, time zones, functions, and cultures [3] Several authors and researchers argue that performance and innovation are strongly dependent on the ability of how organizations are able to work in networks of collaboration [4–8]. Research shows that networks of collaboration positively contribute to gaining competitive advantages [4,6], are a powerful source of innovation [9,10]—especially if they include human diversity [11–14], and increase performance if they are efficiently distributed across organizational functions, geographies, and technical expertise domains [15]. Adding to this, latest research shows that if organizations want to increase the chances of achieving a sustainable competitive advantage, managers should adopt an ambidextrous leadership style [16–19]. This leadership style is characterized by exploiting present conditions by optimizing the current business model´s operation, and simultaneously, exploring opportunities that help redefine the business model by taking pioneering risks. This demands more flexibility, awareness, and insight, and ultimately, working more in networks of collaboration. Indeed, some authors argue that in today´s business landscape, although individual

competency and training are important factors, it is almost always the network factor that is the big key predictor of high performance in organizations and are usually characterized by having broader and diverse problem-solving networks fueled with positive energy [20]. But a broader network does not necessarily mean larger in size. Furthermore, some authors argue that when it comes to effective networks of collaboration, bigger does not always means *better*, but rather, superior quality means better [7,8]. However, creating a broad and diverse problem-solving network usually requires an extra mile from the employees of an organization, essentially because they need to be more flexible, accountable, and empowered, and proactively search and maintain such networks. This also means that more work will be done through informal networks of relationships, removing to a certain extent, the role of the formal organizational structure in several ways [8,21]. In fact, building a broader and more diverse problem-solving network requires a proper organizational structure that enables people to strategically create the necessary connections in an energized way. Usually, due its rigid nature, a formal organizational structure [6] is not able to provide for the needs of building such problem-solving networks [8,22]. However, either under the pressure or when highly motivated to get the work done, employees of an organization naturally engage in informal networks of collaboration, in order to overcome the natural constraints of the formal organizational structure to get the work done [8,22,23]. Very often these emerging informal networks are not ruled by the rational-legal authority system that the formal organizational structure provides based on universalistic principles that are understood to be fair, but rather, by fundamentally unfair and particularistic principles, such as friendship, homophily, propinquity, dependency, trust, and others, which are characterized as personal and social needs of individuals [22]. Therefore, if these emerging informal networks—which are usually hidden behind the organizational formal chart and extremely hard to spot with a naked eye [23]—are not properly managed, they can turn into an issue and strongly hinder the performance and innovation capacity of an organization [8,21]. Research shows that organizational informal networks have a pervasive influence on employees' experience of work, being often critical to how they find information, solve problems, and capitalize on opportunities, and are intimately intertwined with employee satisfaction, well-being, and retention [24]. Ultimately, if these informal networks are not properly managed, they can evolve either into a collaborative overload, or lack or inefficient collaboration patterns [25]. In project environments, the emergence of project informal networks, as an to answer to the day-to-day challenges, occurs at a high speed, and very often is neglected by managers. However, project networks, can be governed and coordinated in very different ways [26]. David Hillson, a renowned name in the field of project risk management, argues that although the number of bodies, standards, and institutes that provide guidance in project management is increasing, projects still fail at an alarming rate [27,28]. In fact, according to the Standish Group CHAOS report-2015, from 2011 to 2018, only about 29% of executed projects had successful outcomes [29,30], according to traditional project success outcome criteria (time, budget, and scope). The same trend was output by the PMI (Project Management Institute) Institute, in the PMI—Pulse of the Profession® 2017 report, where it shows that from 2011 until 2017, projects on average have been completed on time and on budget less than 60% of the time, where drivers such as changes in an organization's priorities, inaccurate requirements gathering, inadequate or poor communication, and team member procrastination, are amongst the most nominated as being responsible for this outcome [31]. Hillson argues that the unsuccessful projects percentage occurs essentially due three major reasons that directly comply with project risk management [27,32]. First, processes need to be consensually aligned regarding approaches and risk management standards. Second, principles need to be redefined in order to remove subjective understandings of what risk and project management really is. Finally, people—projects and risk management are still done by people and not machines. Therefore, people's different cultures, know-hows, skills, informal interactions, roles, and dynamics need to be deeper researched and less neglected [33]. The people aspect has been highlighted in a publication in 2018 at the Harvard Business Review under the name "*Better People Analytics*," where two researchers from the people analytics area concluded that, besides the two traditional people analytics factors traits and states,

a third factor should be considered [34]. This third factor, coined by the researchers as relational data, is the mapping and analysis of the employee´s informal organizational relationships, in six different areas. They are: ideation—the prediction of which employees will come up with good ideas; *influence*—the prediction of which employees will change others' behavior; *efficiency*—the prediction of which teams will complete projects on time; *innovation*—the prediction of which teams will innovate effectively; *silos*—the prediction of whether an organization is siloed by measuring its modularity; and *network vulnerability*—the prediction of which employees the organization cannot afford to lose. Although organizational informal networks should be identified and properly managed, the latest research indicates that organizations alone, are not able to do that by themselves, essentially due the nature of the purpose for which organizations are designed, which is to drive operational efficiencies, by managing, coordinating, and controlling activities [6]. The challenge is then, how can we uncover and properly manage these informal networks of collaboration? The answer can be found on the application of social network analysis in organizations. In a nutshell, social network analysis (SNA) is the process of studying social structures, by usually analyzing social dichotomous data with a variety of measures developed based on graph theory that helps to explain how those social structures evolve trough time, and how they impact the environments where they exist [35,36]. Social network analysis theory can play a very important role in bringing light to the social capital challenges [37]. Moved by the impact both positive and negative that informal organizational networks may have in an organization's performance and innovation capacities, researches, institutes and consulting organizations have been incorporating what some coined as people risk management models into their traditional risk management processes [38–40]. At the same time, specialized people risk literature has been outputting latest trends and developments in this area, which includes but is not limited subjects such as to talent shortages and retention, incompetence, innovation, working in networks, collective and individual performance, cultural fit, values, unethical behavior, low morale, employee wellness, and noncompliance with industry, and fraud [41,42]. In this work, a further contribution to this trendy people risk field in the area of project management is given, by proposing a heuristic model to identify the extent to which the dynamic interactions of project people throughout all the phases of a project lifecycle, influence a project outcome. The proposed model will analyze and quantitatively measure the project-related information that *flows* across the naturally emerging project´s informal network, throughout all the phases of a project lifecycle. The proposed model is named the project outcome likelihood model (POL) and was developed based on three scientific fields (Figure 1). They are: project management theory—which contributes with the terminology used in project management, and the structure of a project lifecycle; social network analysis theory—which contributes with the tools and techniques to uncover and measure the dynamic interactions of project people throughout a project lifecycle; and risk management theory—which contributes with the risk identification and treatment process and framework.

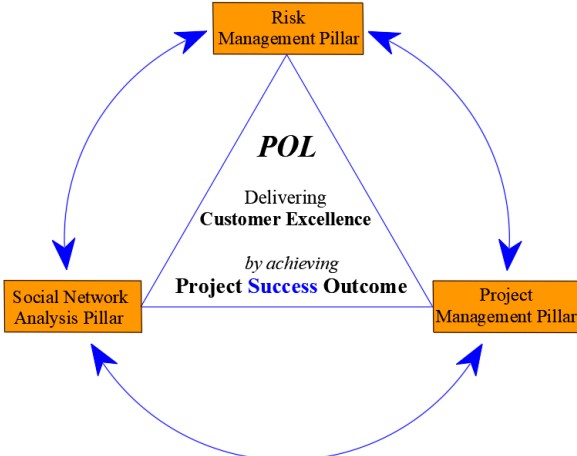

**Figure 1.** Three scientific pillars that form the basis to the development of the POL model.

## 2. Literature Review

### 2.1. History and Evolution of Social Network Analysis

Since the publication of the book "*Who Shall Survive*" by the Romanian-American psychiatrist, psychosociologist, and educator Jacob Levy Moreno (1889-1974) in 1934, research and development in the field of social network analysis has been exponentially increasing in a wide range of different areas [43–49]. It spans social and behavioral sciences [50], agriculture [51] health care [52] environment [53], law, national safety, criminology and terrorism [54], political science [55], organizational science industry, management and leadership [56], communication, and learning and media [57], just to name a few. J. Clyde Mitchell defined social network analysis as, "A specific set of linkages among a defined set of persons, with the additional property that the characteristics of these linkages as a whole may be used to interpret the social behavior of the persons involved" [58]. Although the application of the graph theory has been around for more than two centuries, it was only when it started to be applied to study social structures that it exponentially rose in popularity, essentially motivated by the desire to understand the extent to which people´s behaviors and relationships influence others and outcomes, usually translated into performance, innovation, social cohesion, information diffusion, and so on [49,59,60]. Indeed, these types of relationships are complex by nature, and cannot be completely explained trough traditional individualistic social theory and data analysis methods, but rather by methods that are embedded in sociology that consider the individual´s social context in the process of making choices [61]. It is exactly at this point that social network analysis plays a critical role, by providing valuable insight [61,62]. One of the very first publications, arguing the benefits of applying social network analysis in organizations, was published in 1979 by network researchers Noel Tichy, Michael L. Tushman, and Charles Fombrun [63]. In their work, they suggested that by applying social network analysis to organizations, significant advances could be made in organizational theory and research, because this approach would facilitate the comparative analysis of organizations as well as the comparison of subunits within an organization. Furthermore, they outlined that the benefits of applying SNA were substantial and appeared to far outweigh the costs. In fact, in a comparative scenario between two organizations, where, for example, it is assumed that the performance and innovation levels of both are direct consequences of the dynamics of their informal networks, only when those dynamics become quantitatively measured (which can be done by applying SNA) can the comparison be done.

*2.2. Social Network Analysis and Project Management*

2.2.1. Project Management Challenges

A project is a temporary endeavor with a defined beginning and end, designed to create a unique product or service, and is expected be properly managed by the application of knowledge, skills, tools, and techniques to project activities to meet the project requirements, throughout all the different project phases that comprise a project lifecycle [64]. Due lack of consensus regarding how many phases a project should be divided into, a common approach is that the number of project phases is determined by the project team, and/or the type of the project. Project management challenges, usually called project risks (essentially threats, rather than opportunities), represent essentially the likelihood of not delivering planned project activities within the constraints (usually, scope, quality, schedule, costs, and resources [64]) associated to a project. According to David Hillson, project risk is the uncertainty that matters [65] and aims to separate what a *real* project risk is, from what a *not real* project risk is. Hillson suggests four types of uncertainties that could affect a project's ability to achieve its objectives, regardless of a project phase [65]. They are: 1—Event risk: Possible future events, sometimes called "stochastic uncertainty" or "event risks." An event risk is something that has not yet happened and may not happen at all, but if it does happen, it has an impact on one or more project objectives; most of the risk identified at the project risk register is of this type, and a set of well-established techniques for identifying, assessing, and managing them already exists. 2—Variability risk (also called "aleatoric uncertainty"): They are a set number of possible known outcomes, among which it is not known which one will really occur. These types of project risks cannot be managed by applying standard risk process procedures, but rather advanced analysis models, such as the Monte Carlo simulation. 3—Ambiguity risks (also known as "epistemic uncertainties"): Uncertainties arising from lack of knowledge or understanding. Could be called know-how and know-what risks. They comprise the use of new technology, market conditions, competitor capability or intentions, and so on. These can be managed by learning from experience of others—lessons learned—and by prototyping and simulating before taking real action. The proposed model in this work can be seen as a framework to manage ambiguous risk types, once it aims to identify critical success factors from past events (lessons learned), and use identified critical success factors to guide and predict (simulations) a future outcome of an ongoing project. 4—Emergent risks: They emerge from people´s blind-spots. Usually called "ontological uncertainty," they are more commonly known as "black swans." These risks are unable to be seen because they are outside a person´s experience or mindset, so one does not know that he should be looking for them at all. Usually they arise from game-changers and paradigm shifters, such as the release of disruptive inventions or products, or the use of cross-over technology from previously unforeseen sources. To manage them, contingency seems to be the key word, sometimes defined as "the capacity to maintain core purpose and integrity in the face of external or internal shock and change". Pinto and Slevin, 1988 [66], uncovered a set of ten critical success factors that, contrary to Hillson, change importance according to a project phase. These critical success factors are considered major project risks that if not properly managed, will hinder the chances of a success outcome. They are: (1) project mission not being properly defined, (2) lack of top management support, (3) non-detailed project schedule, (4) poor client consultation, (5) lack of necessary and proper technology and expertise, (6) poor team skills and experience, (7) ambiguous client acceptance, (8) lack of proper monitoring and feedback of project activities, (9) poor or lack of proper communication, and (10) non-readiness to handle excepted crises and deviations from plan (in other words, lack of a contingency plan). It can be said that regardless of a project phase, the above-mentioned risks and critical success factors, if not properly managed, will damage the likelihood of achieving a successful project outcome.

2.2.2. Application of Social Network Analysis in Project Management

The application of social network analysis has been in the last few years extending into the area of project management, although this adoption remains so far at a very initial stage [67,68]. In project

environment, objectives such, as but not limited to, understanding and measure knowledge transfer, consensus building, and the identification of critical success factors—regarding the dynamics of project informal networks—which may contribute to a success project outcome, have been priorities of the application of social network analysis [69,70]. Regarding the identification of critical project success factors though, still without applying social network analysis, it has been a present preoccupation within the most recent years. One of the very first notable works regarding the identification of critical success factors in project management was published by Pinto and Slevin in 1988. By surveying circa 400 project managers from very different industries areas, asking them what major reasons that lead to success and unsuccess projects allowed them to identify a set of common critical factors that were indicated as responsible for a successful project outcome. Furthermore, they identified that those critical factors were changing the importance degree, function of the project phase [66]. They identified a *top ten set* of critical factors. Three of them are related to how project people work in networks of collaboration throughout the different phases of a project lifecycle. They are top management support, client consultation, and communicating network. The findings made by Pinto and Slevin in 1988, were revalidated by research conducted in 2005 and 2012 [71,72]. This study contributed to triggering the interest of network and project researchers that soon brought social network analysis into the project management field. They are trying to provide valuable insight into some of these critical factors identified by Pinto and Slevin. In a publication at the Harvard Business Review (HBR) in 1993, professor David Krackhardt, and Jeffrey R. Hanson, highlighted the importance of managers tapping their informal organizational networks as being a key contributor for success, essentially by mapping three types of networks: the *advice* network—which reveals the people to whom others turn to get work done; the *trust* network—which uncovers who shares delicate information; and the communication network—which shows who talks to whom about work-related matters [8]. These three networks would then be mapped with the employee´s relational information collected through surveys. They argued that this approach would get to the roots of many organizational problems. In 2001 Stephen Mead conducted one of the first, and top ten most-cited case studies ever [73], wherein he applied social network analysis in a project environment to visualize project teams [74]. By the application of SNA, Mead identified and analyzed an informal project stakeholder´s communication network. Mead identified isolated and central stakeholders regarding the informal communication network and elaborated a corrective plan in order to improve the performances of those that were isolated. In the latest 30 years, remarkable research has been done by Professor Rob Cross, a renowned researcher and developer who applies social network analysis to study organizations. In one of his works, "*The Hidden Power of Social Networks,*" published in 2004, he collects a set of ten-year research cases in the study of organizations, and he highlights the many benefits of the application of SNA in organizations, especially in project environments [23]. Cross found that in every organization, there is an informal organizational network that is responsible for how the work is done. Furthermore, he argues that in every informal organizational network, there are a set of common actors who are responsible for most dynamics of an organization. These actors include: the central connector, the boundary spanner, the information broker, the peripheral expert, the peripheral intentional, and the energizer. These names were coined to functionally describe their positions in the informal network structure [21,23]. An application of social network analysis in the health projects environment is illustrated by a study conducted by the U.S. department of health and human services, where SNA methods and measures were applied to understand the multidimensional determinants and complexity of tobacco use [75]. Prell et al. (2009) applied SNA to analyze stakeholder networks in a natural resource management, where they identified which individuals and categories of stakeholder played more central roles in the project network and which were more peripheral, leading this information to guide stakeholder selection. Another notable work, regarding the application of social network analysis in projects, was published in one of the most credited project management institutes [64] in 2012; it details the importance of four key subjects of social network analysis, showing that those key subjects are directly related to project management performance [76]. They are centrality, structural holes, boundary management,

and tie strength. The other latest studies regarding the application of social network analysis in project environments, go from the development of models based on SNA to analyze communication, collaboration, and knowledge networks in project meta-networks [77], to the analysis of mega projects networks, from the perspectives of different important project stakeholders that help to develop proper long-term project governance policies [78,79]. In 2017, Mok et al. (2017) [80] applied SNA basic network centrality measures to identify key challenges in major engineering projects (MEPs) based on interdependencies between stakeholder concerns, resulting in the identification of a set of key challenges that that occur in such MEPs, and helped to properly develop a set of recommendations to alleviate those challenges, which could be used in future MEPs. Yu et al. (2017) [81], used social network analysis to investigate social risks related to housing demolition, from a stakeholder perspective in China. A recent work in the field of organizations, developed by Michael Arena, Chief Talent Officer for General Motors, concluded that, after years of investigation in several organizations, successful organizations operate in a networked way, enjoying what they called an *adaptive space*, which enables a proper connection between the operational and entrepreneurial pockets of an organization in a *virtual* space—adaptive space—where employees explore new ideas, and empower the most creative people to spread their ideas across the organization. This adaptive space is built, managed, and maintained using social network analysis [6]. Arena argues that this *adaptive space* enables organizations to work in a more agile way, which ultimately contributes to outperforming the competition in a disruptive way. However, research and development in the area of SNA extends to other areas, as does the example of the research done by professor Eric Xing, where he conducted an investigation on dynamic predictive models, built on social network analysis theory, which have as objectives, forecasting how people will interact and react when facing different future events [82]. Such predictive models, integrated into the risk management process of an organization, may strongly help organizations to better, and more accurately prepare projects or operations, by estimating with a higher certainty degree the evolution of future events, and thus, prepare proper plans to respond to those events. In the field of project management, such dynamic predictive models will bring critical help, namely, at the stage of choosing elements to form a project team that better will adapt to certain upcoming project events, by forging the necessary hard and soft skills that are needed to face future events. Finally, an indicator of the growing trends regarding the importance of the application of social network analysis in organizations to analyze the influence that the informal organizational network may have on performance, innovation, employee retention, and so on, is visible in renowned consultancy organizations such as, but not limited to, Deloitt [40] and Mckinsey [83], as they continuously keep integrating social network analysis in their people analytics toolkits.

### 2.2.3. The Importance of Centrality in Project Management

Centrality in a social network refers to the structural attribute of an actor—where an actor is located within the structure of a social network—and not to the actor´s own inherent attributes such as age, tender, expertise, and so on. Throughout years of research, leading social network researchers have been arguing that actor centrality can be a measure importance, influence, prestige, control, and prominence [84–86], and these can be quantitatively measured by applying graph theory centrality metrics, such as *degree*, *betweenness*, and *closeness* [43]. According to Freeman (1979), for each of these metrics a respective social direct implication exists as follows: activity (degree can be an index of potential for the network´s activity), control (betweenness is an index of communication control by serving as bridge between two different subgroups of an network), and independence (closeness is an index of the potential independence from network control) respectively. In other words, network centrality can be seen as a source of informal power in a network. In this context, informal power derives from the advantageous position of an actor in the network regarding interaction patterns, such as communication, collaboration, information exchange, and so on, whereas formal power is defined by the position of an actor in the organizational formal chart [87]. It can be argued that in an organizational context, people occupying those three mentioned social network structure positions

(degree, betweenness, and closeness), have a greater responsibility in the maintenance and coordination of the whole network. As previously shown, several studies have focused on the centrality subject in informal networks as a special location within a structure of an informal project network [21,23,76]. In fact, further research has been showing that centrality in informal networks plays a key role regarding project coordination and decision-making. Liaquat et al. (2006) [88] applied network analysis to explore the correlation between actor centrality and project-based coordination, concluding that in projects, actors who are central and well connected in informal project networks are able to exercise greater coordination within the network structure. Dogan et al. (2014) [89] reinforced the idea of the importance of centrality in informal networks, as he applied social network analysis to measure coordination performance in building and construction projects, concluding that coordination scores are highly correlated with centrality indices. Wen et al. (2018) [90] applied social network analysis to investigate the determinants of timely decision-making from the perspective of collaboration network dynamics and concluded that network tie strength (familiarity) and network position strength (centrality) have a positive effect in a real-world project decision setting.

## 3. Model Development and Implementation

### 3.1. The Proposed Model in a Nutshell

The model proposed in this work is called project outcome likelihood (*POL*) method, and aims to provide valuable insights that may contribute to answering the following question: to which extent, do the dynamic interactions of project people throughout all the different phases of a project lifecycle, influence a project outcome? In a nutshell, to provide valuable insights that may contribute to answer the question presented above, the proposed model is designed to identify repeatable behavioral patterns that are associated to a certain project outcome type (success or failure). In other words, the model will analyze how project people behave throughout the project lifecycle of a successful delivered project, and how project people behave throughout a project lifecycle of an unsuccessful delivered project. If the model finds that in projects that were successfully delivered, project people clearly behaved differently—regarding five different global collaboration types (**5-GCT**)—than in projects that were unsuccessfully delivered, then the answer to the question above presented, is found. Furthermore, if indeed different behaviors are found, they are considered project critical factors, with obvious focus on what the project critical success factors are. Those different behaviors will be identified by analyzing and measuring information arising from project meetings, emails, and questionnaires through the application of social network analysis. This information mirrors the dynamic interactions of project people throughout all the different phases of a project lifecycle, and is used to characterize the five different global collaboration types, which are (Table 1): (a) communication (b) internal and cross boundaries-collaboration, (c) know-how and power sharing, (d) clustering (variability effect), and (e) teamwork efficiency. The complete framework is illustrated in Figure 2. After the identification of the critical success factors, the model proposes a framework to monitor the evolution of an ongoing project by comparing an actual state against a desired state based on the identified critical success factors. This framework is illustrated in Figure 3.

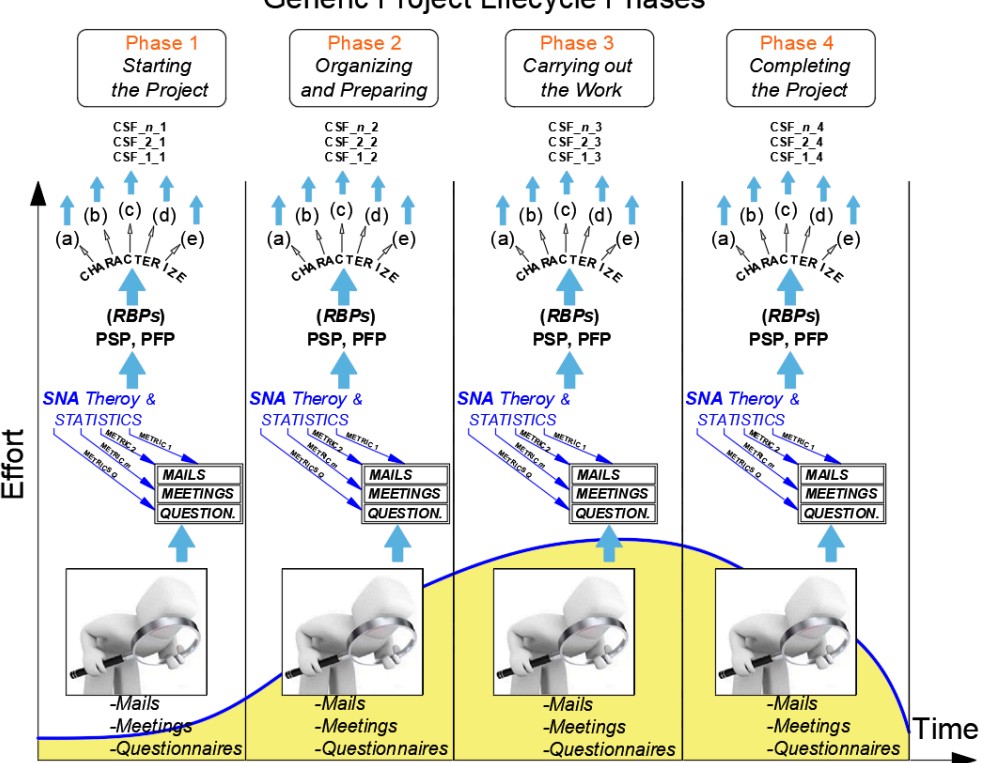

**Figure 2.** POL model: Part 1's process for success and failure project outcomes. Source: adapted PMI, 2017.

**Table 1.** The five global collaboration types (5-GCT) of the POL model.

| | |
|---|---|
| (a) Communication and Insight | Description and Objective: How does important project stakeholders (Project Managers and Experts for example), are conducting the project-phase global communication, and the impact for a certain project outcome? How is the feedback level between the different teams that work in networks of collaboration?<br>Regarding Meetings: How the presence of those important stakeholders in project Meetings, triggers communication and insight on what is ongoing throughout the different projects of a project lifecycle, namely at the transitional phase of the different project phases.<br>Regarding Mails: How does the internal and external mail communication is being made regarding cohesion? How is the feedback rate? How is the feedback rate regarding mails communication network? |
| (b) Internal and Cross Boundaries-Collaboration | Description and Objective: To which extent is one Team (service provider for example), more dependent, or less dependent on project-related information provided from other Team (customer for example), and the impact for a certain project outcome?<br>Regarding Mails: How is the volume of mail communication seeking and providing project-related information between any two Teams (internal to Team A—sub-team of Team A, or cross boundaries, between any two different Teams, as Team A and Team B? |
| (c) Know-how sharing and Power | Description and Objective: To which extent, does the project-related information, provided by one Team (service provider), or other Team (customer), is recognized as important and decisive, and the impact for a certain project outcome?<br>Regarding Questionnaires: Apply a SNA assessment, to an Outsourced organization, to find out on which side (service provider, or customer), lays the major know-how and decision-making power, on site. |
| (d) Clustering (variability effect—*PSNVar*) | Description and Objective: To which extent, does the variability (changing the project team set) of the project social network cohesion, across all phases of a project lifecycle, contributes for a certain project outcome?<br>Regarding Meetings: How frequent do changes on the project team set occur, across the different phases of a project outcome, and how is reflected in project people social cohesion? |
| (e) Teamwork efficiency | Description and Objective: To which extent, does a project outcome is associated to the speed of feedbacking project information-related between project teams?<br>Regarding Mails: To which extent does the Email Feedback Speed, when answering a question or providing project information-related between different project teams, impacts project outcome? |

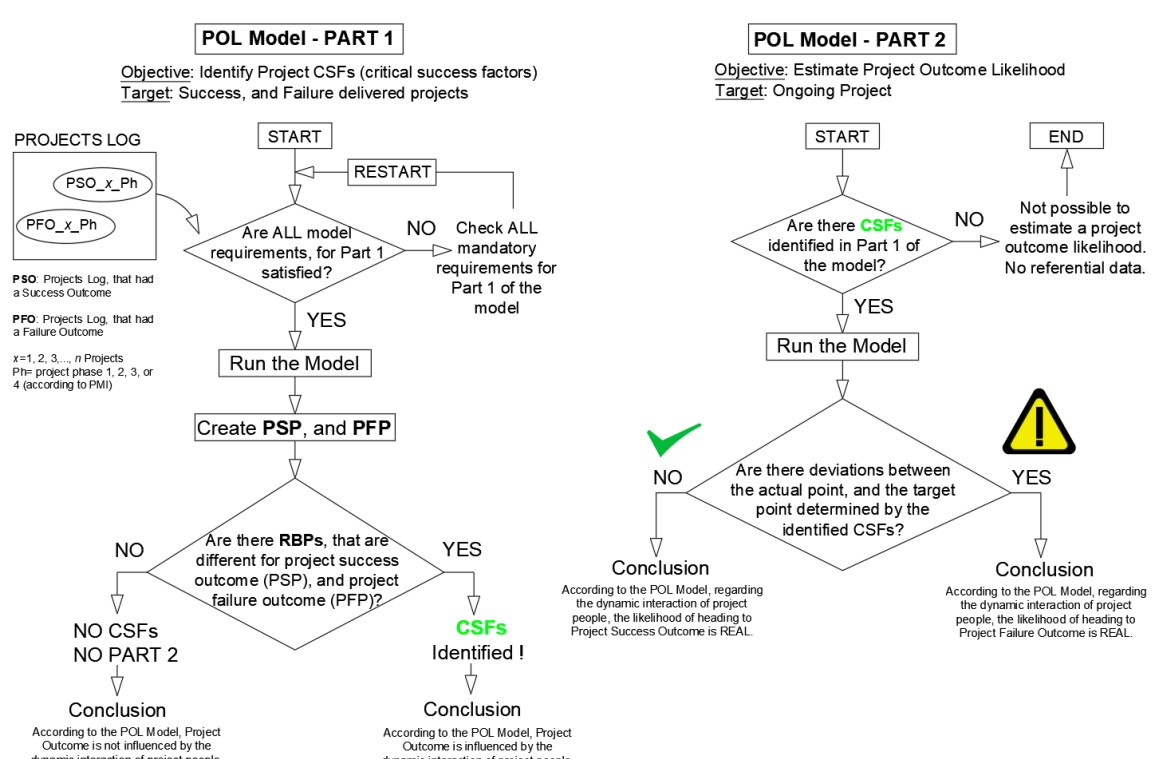

**Figure 3.** Model framework—Part 1 and Part 2.

*3.2. Model Key Concepts*

### 3.2.1. Project Definition

Before drilling down on how the model is built and operates, important subjects need to be introduced. The proposed model will analyze projects that were successfully and unsuccessfully delivered. First, a project is to be defined, according to the PMI, as a temporary endeavor undertaken to create a unique product, service, or result [64].

### 3.2.2. Project Outcome

Regarding how a project is delivered, there are only two possible types of outcomes—successful and unsuccessful. These two types of project outcomes—successful and unsuccessful, are the only ones accepted by the proposed model, and the criterion that dictates both types is given by the PMI [64]. It essentially says that a successfully delivered project is a project that was delivered on time, within the agreed scope, budget, and the quality.

### 3.2.3. Number of Projects

The proposed model does not preview a minimum or maximum recommendable number of projects that were successfully and unsuccessfully delivered to be analyzed. There is a minimum number of projects that needs to be used as input to the model so that it can function. This number is two: one successful and one unsuccessful delivered project. However, in order to obtain more significant results that better represent, to a certain extent, the overall working culture of a service provider-organization, while delivering projects for one or multiple customers, a substantial number of projects is recommended. It would not be an exaggeration to say the more projects, the better.

### 3.2.4. Project People

Project people are any people who directly or indirectly participated in a project, throughout its lifecycle. These are the internal stakeholders from both interacting parts—service provider and the customer and are contractually engaged for the demand/supply of resources, services, and/or end products in project delivery [64,91]. In other words, project people are any people who have participated in project meetings, email project information-related exchanges, and questionnaires. Nevertheless, as the model previews only the study of the dynamic interaction between two different teams (Team A—service provider and Team B—customer), across a project lifecycle, all project people are expected to belong to one of the two sides—Team A or Team B, regardless of the type of relationship with one of the sides—outsourcing, consultant, or other. In a case where there was participation of project people who did not belong to either Team A or Team B, they were not considered by the proposed model. This is the case regarding, for example, external consultants or audit-teams that were both agreed on by Team A and by Team B to come into play across a project lifecycle, usually playing a neutral role characterized essentially by advice and guidance.

### 3.2.5. Project People-Roles

Across a project lifecycle, it is usually expected that several different project people-roles, with their own and specific responsibilities, take part on project activities. Also called important stakeholders, project people-roles for the proposed model are: the project manager(s), engineer(s), expert(s), other(s), and outsourcers. Others include all those project roles that have not been previously mentioned.

### 3.2.6. Project Phases and Lifecycle

Any project that has been successfully or unsuccessfully delivered, has a finite number of project-phases. The number of phases is given by the PMI standard, where four generic phases are previewed. They are: starting the project, organizing and preparing, carrying out the work, and ending

the project [64]. The sum of all project phases of a project is the so-called ***project lifecycle***. The sum of all project phases of a project is the so-called project lifecycle.

### 3.2.7. Dynamic Interaction

The dynamic interactions of project people (DIPP), refer to how project people communicate, direct, and cross boundaries to collaborate—share know-how, exert power, cluster (create and lose relationships), and provide advice or help—within the project informal network, and they are analyzed by measuring the degree of meeting participation by project people, the rate and intensity of emailing project information-related material, and the degree of importance and influence of some project people over other project people, measured though information collected in questionnaires.

### 3.2.8. Project Informal Network

The project informal network, or project social network (PSN) refers to all the project people's dynamic interactions that occur out of and inside the pre-defined formal structure format. This means that, for example, in the email project information-related exchange network, the communication between a project people A and a project people B, will be analyzed and considered part of the project informal network, even when A is the direct superior (according to the project formal chart) of B. The same principle is applied if project people A and project people B have no direct formal dependency relation (superior nor subordinate); still the email-based project information-related exchange network will be analyzed and considered part of the project informal network.

### *3.3. Model Function Principles*

As previously said, the proposed model will look for repeatable behavioral patterns (RBP) regarding the dynamic interactions of project people across the different phases of a project lifecycle, for both project outcomes. If the model identifies unique RBPs that are associated with a certain project outcome (success or failure), they are classified as critical factors. Unique RBPs mean that there are things that repeat themselves only in one of the two possible project outcomes. By being unique, they are immediately considered project critical success factors by opposition. To better illustrate the above description, a theoretical example will be explained using Figure 4 as support. Figure 4 presents the lifecycles of two real delivered projects—Project 1 and Project 2—which were delivered to a Customer C (not present at the Figure 4) and accomplished by the Organizations A and B (both present at Figure 4). Both Projects 1 and 2 comprise the development and implementation of technical solutions in the food and beverage industry. Essentially, Customer C, a market leader in the food and beverage industry, invited first, several organizations that deliver projects in the mentioned area, such as Organizations A and B, to show their plans/ideas, and to develop and commercialize a new product; and second, to create two project proposals that meet its needs: namely, building two new production lines. Organizations A and B won both project proposals and delivered both projects, 1 and 2. Project 1 and Project 2 occurred at different points in time, but within the same year in Europe. Both projects were delivered across the year of 2019, with the average duration of 5 months each. Organization A delivered the engineering and mechanical installation parts, and Organization B delivered the automation and programming services. Project 1 had a budget of circa 8 M€, and Project 2 had a budget of circa 5 M€. For both projects, 1 and 2, the same formal structure was used regarding Organizations A and B respectively, according to Figure 4. Due to protection reasons, further project information, and the identities of Customer C and Organizations A and B, will not be disclosed in this work. Employees (project people) from Organization A are represented with the black color (Team A), and from the Organization B, with the green color (Team B). At Figure 4, the project people's locations within the project's formal structure are also illustrated, which is valid for both Projects 1 and 2, where for example, for Team A, PP1 is the project manager; PP2, PP3, PP4, and PP5 are the direct subordinates (functional managers); and PP6, PP7, and PP8 are project staff (engineers, programmers, and so on). The project people represented in the formal chart are those project people that were assigned at the

very beginning of Phase 1 to deliver the project throughout all the project phases. Concretely, Figure 4 represents the collaboration network between Organizations A and B, while delivering Projects 1 and 2 to a Customer C. The blue line (project planned curve) represents how the projects were planned to be delivered. The grey line represents how the projects were delivered. Project 1 was successfully delivered, and Project 2 was unsuccessfully delivered. For this illustrative case, and according to the PMI, a successfully delivered project is a project delivered on time, within agreed scope, budget, and quality [64]. On the other hand, an unsuccessfully delivered project is a project where at least one of the mentioned constraints (time, scope, budget, or quality) was not delivered according to the plan. Both projects have four different phases: Phase 1, 2, 3, and 4 throughout their lifecycles. In each of the four phases of both projects, the light blue lines represent the email communication networks between the project people. Such networks mirror the direct communication (emails sent directly to, or received directed from) patterns; the results of the analysis of all project-information-related emails at any given project phase. For example, after consulting the email exchange logs for Project 1 at Phase 1, there was direct communication between project person 4 and project person 5. This means that PP4 sent and or received an email from PP5, and vice versa, at any given time within the Phase 1 of Project 1. On the other hand, for example, on Project 2 at Phase 1, PP3 and PP5 never had direct contact regarding email exchange. Each light blue line between any two project people represents an email communication channel regardless of the number of emails exchanged. When analyzing how the email communication occurred at the first two phases (Phase 1, and Phase 2) of both projects, it is clear that for the project that was successfully delivered (Project 1), the email communication network was by far denser than the email communication network in Project 2.

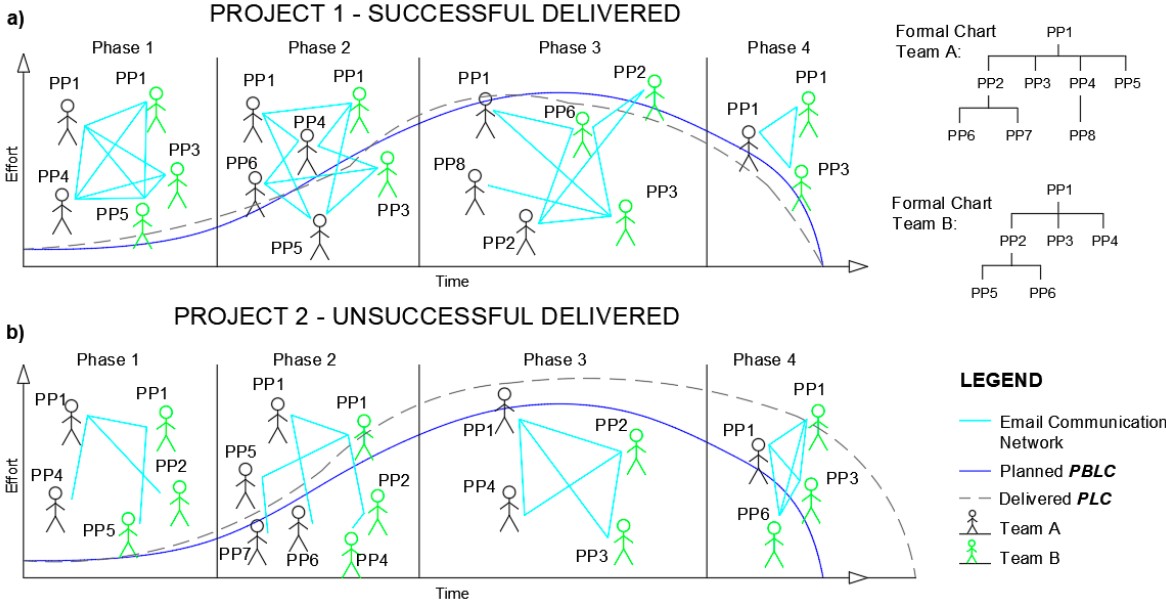

**Figure 4.** Project lifecycles for Project 1 and Project 2.

For these two phases, and not considering any other factors, it can be concluded that a denser email communication network in Phase 1 of a project is correlated to a project success outcome. Therefore, the email communication network is a critical success factor, and project success outcome is correlated with a dense email communication network at Phase 1 of a project. However, one still needs to define what dense and sparse email communication networks are. This means that they need to be quantified. It is now that the social network analysis provides critical help. SNA uses the graph theory that can be used to characterize and measure a social structure. The constellation that results from linking project people through emails sent and received perfectly mirrors a typical graph structure. Therefore, the direct application of graph theory metrics is adequate. For this case a centrality measure—the

*density* [43] metric—will be used to quantify what a dense and a sparse communication network is. The density is the ratio of existing links between project people inside each phase of each project lifecycle, to the maximum possible number of links (when everybody is linked) between project people, and is given by Equation (1):

$$d = \frac{2L}{n(n-1)} \tag{1}$$

where:

$L$ = number of existing lines between project people;
$n$ = total number of project people;

Applying Equation (1) for Phase 1 of Project 1:

$$d = \frac{18}{5(5-1)} = 90\% \tag{2}$$

Applying Equation (1) for Phase 2 of Project 1:

$$d = \frac{20}{6(6-1)} = 67\% \tag{3}$$

Applying Equation (1) for Phase 3 of Project 1:

$$d = \frac{16}{6(6-1)} = 53\% \tag{4}$$

Applying Equation (1) for Phase 1 of Project 2:

$$d = \frac{8}{5(5-1)} = 40\% \tag{5}$$

Applying Equation (1) for Phase 2 of Project 2:

$$d = \frac{12}{7(7-1)} = 29\% \tag{6}$$

Applying Equation (1) for Phase 3 of Project 2:

$$d = \frac{10}{4(4-1)} = 83\% \tag{7}$$

After analyzing the results, it can be concluded that the density value, for example, for Project 1 at Phase 1 is 90%, and for Project 2 at Phase 1 it is 40%. This means that, at Project 1 in Phase 1, there was 50% more direct coverage regarding the email communication network, than for Project 2 in the same phase. In other words, this means that for Project 1 at Phase 1, almost everybody directly communicated with everybody through the email network, at a certain point in time. There may be several interpretations for this. One, for example, is that according to the legend, PP1 for both Projects 1 and 2 is the respective project manager for Team A or B, and PP4 is a direct subordinate, it can be concluded that the email communication network for Project 1 follows a more informal pattern than the one for Project 2. This is because for Project 1 in Phase 1, one project manager subordinate (PP4), had direct contact with all the other project people, in opposition to the PP4 at Project 2, where the email communication network follows a more formal pattern; the email communication between Organization A and Organization B is exclusively done through the project manager of Organization A. For example, in Project 1 at Phase 2, the direct email communication channels drop to 67%, which means that from all the possible direct email communication channels, only 67% of them exist. Now when analyzing Phase 4 of both Projects 1 and 2, it can be seen by the naked eye that all the project

people are directly connected through an email communication channel. In fact, when applying Equation (1) for both projects at Phase 4:

Applying Equation (1) for Phase 4 of Project 1:

$$d = \frac{6}{3(3-1)} = 100\% \tag{8}$$

Applying Equation (1) for Phase 4 of Project 2:

$$d = \frac{12}{4(4-1)} = 100\% \tag{9}$$

In this case, for the Phase 4 of both projects, there is no difference between the density results on the both projects. However, Project 1 was successfully delivered, and Project 2 was unsuccessfully delivered. Therefore, it can be concluded that, the email communication network for Phase 4 of both projects is no longer considered a critical success factor, because there is absolutely no difference between the results for both projects, regarding the density metric. These results are in line with findings from Pinto and Slevin in 1988, as they found that project critical success factors change in degree of importance or even disappear, at the function of project phase. Therefore, for Phase 4, there is a need for another SNA metric that may identify a pattern regarding the dynamic interaction of project people that can be correlated to a certain project outcome. Finally, when comparing Project 1 with Project 2, regarding the email communication network, it can be concluded that for Phases 1, 2, and 3, it is denser in Project 1 than in Project 2. This means that there is a repeatable behavior pattern (RBP) at phases 1, 2, and 3, of Project 1 (successful outcome), which is characterized by having a dense email communication network, of values 90%, 67%, and 53% respectively.

### 3.4. Project Success Profile and Project Failure Profile

Now, let us assume that the collaboration between Organization A and Organization B had successfully delivered 20 projects to the imaginary Customer C and other 20 unsuccessfully delivered projects to the same imaginary Customer C within the latest year (making 40 projects in total), and that the project lifecycle of Project 1 in Figure 4 no longer represents the project lifecycle of one single successfully delivered project, but rather the averaged lifecycle of all those 20 successfully delivered projects within the latest year. The same goes for Project 2 in Figure 4, where it now represents the averaged lifecycle of all those 20 unsuccessfully delivered projects within the latest year. In this case, the light blue lines in the lifecycles of Figure 4 now represent the average repeatable behavioral patterns regarding the email communication network. That means that on average, all the 20 successfully delivered projects had at Phases 1, 2, and 3, a denser email communication network than for the 20 unsuccessfully delivered projects for the same phases. The average results of the email communication network represent one aspect of the so-called project success profile (**PSP**) and project failure profile (**PFP**), for successfully and unsuccessfully delivered projects respectively. A project's success or failure profile represents all the different SNA metrics that were used to analyze and quantitatively measure the dynamic interaction of project people across a project lifecycle. Until now, only one metric has been used to characterize and measure one type of dynamic interaction that occurred at the project informal network, which is the email communication network. Concluding, a project profile is the collection of all averaged SNA-metrics results used to analyze and measure each individual project from the set of projects that were successfully and successfully delivered. Furthermore, the creation of a project success or failure profile is only meaningful if more than one project has been successfully and unsuccessfully delivered. Then, the process of identifying the critical success factors follows, by adopting the same approach as was the case for the density previously explained.

### 3.5. POL Model Application Span

The proposed model is not limited to any certain fixed number of ***project phases***, as we propose in this work, four generic phases recommended by the PMI. The proposed model is designed to applicable regardless of ***project size and complexity***. The ***project people-roles*** are not limited to those previous mentioned at *3.2.4*, as long they are well defined for both project success and project failure outcomes. Finally, the project Teams A and B that are to be analyzed by the model, are necessarily limited to a relationship service provider—Customer as it is proposed in the present work. This means that, regarding ***project teams***, Teams A and B, can be any two different teams in an internal organizational context (between any two business units for example), or an external organizational context (between any two different organizations) that delivers, or/and requests projects. However, the analysis must be always done between any given two teams; for example, a Team A and a Team B, in every project phase, in order to be possible to quantitatively measure. These in turn, may be agglomerated, or delayered into several other "sub" teams. For example, if in a project there is a service provider (Team A), a customer (Team B), and a supplier (Team C), and so on, the model will analyze the dynamic interactions between A and B, A and C, and B and C. In other words, the analysis is always to be done between any two different entities, internal—(if, for example, they are business units, or departments of a given mother organization), and external (if they are different organizations; for example, supplier and customer)—that participate across a project lifecycle's activities/tasks.

### 3.6. POL Model Part 1 and Part 2

Until now, the model only analyzed a set of delivered successful and unsuccessful projects looking for unique RBPs from both project outcomes, in order to identify critical success factors. This is called Part 1 of the model, which is characterized by the identification of project-phase critical success factors. Once critical success factors have been identified, they are to be used as guidance for an ongoing project. This is Part 2 of the model, which is characterized by estimating an ongoing project outcome. This Part 2 of the model can only be executed if Part 1 of the model has been done beforehand and critical success factors have been identified. Otherwise, Part 2 of the model is excluded. Essentially, in Part 2 of the model, an *actual state* of an ongoing project is to be compared to a *desired state* of an ongoing project. The proposed model will then be used to identify whether there are any deviations between the *actual state* and the *desired state*, regarding the dynamic interactions of project people across a project lifecycle. A framework for both parts of the proposed model is illustrated in Figure 3. In order to understand how the proposed model functions, Figures 3 and 4 must be simultaneously interpreted.

For the POL model's Part 1 (Figure 3), required project related-data according to Tables 2 and 3, arriving from number of selected successfully (PSO) and unsuccessfully (PFO) delivered projects will be analyzed and measured (for all the different project phases) by the application of metrics based on social network analysis and statistics (Figure 2). After all selected projects have been analyzed, an averaged profile characterizing all the successful (PSP) and unsuccessful (PFP) delivered projects will be created (Figures 2 and 3). The content in each of the created project profiles is an averaged result, reflecting the repeatable behavioral pattern (RBPs), regarding how project people dynamically interacted throughout the different phases of the selected project lifecycles for both successfully and unsuccessfully delivered projects (Figure 2, Figure 3). It follows a comparison process between the RBPs that characterize the PFP and the PSP. Throughout this process, if crystal clear differences (opposite results regarding PSP and PFP) are identified, then project critical success factor(s) have been identified (Figures 2 and 3). At this point, if project critical success factor has been identified, the characterization of the five global collaboration types is concluded. If critical success factors have not been identified, then it is to be concluded that according to the proposed model, the dynamic interactions of project people across the different phases of the analyzed project lifecycles do not influences a project outcome (Figure 3). In the POL model, Part 2 (Figure 3), if critical success factors have been identified in Part 1 of the proposed model, then for an ongoing project they may be used as guidance regarding the dynamic interactions of project people for the actual project phase of the ongoing project. However,

functions of the metrics to be applied at Part 1 need to be run to output the quantitative results that enable one to compare the actual status of the ongoing project with the *should be* status according to the identified critical success factors. If there is a deviation between the actual status and the *should be status*, then it is to be concluded that according to the proposed model, the likelihood of heading to project failure outcome is real. The contrary is then also true. The outcome likelihood will then be estimated by applying a simple rule based on the highest percentage of metrics indicating success or failure outcome. In other words, the more metric results—for an ongoing project—are aligned with the critical success factors identified at Part 1 of the model, the highest is the success outcome likelihood of the ongoing project phase (Part 2 of the Model). Nevertheless, the present work does not aim to detailed explain the process of estimating a project outcome likelihood.

**Table 2.** Requirements for Part 1 and Part 2 of the POL model.

| Part 1 | Part 2 |
|---|---|
| Objective: Identify project-phase critical success factors. | Objective: Estimate an outcome likelihood for an ongoing project-phase. |
| Method: Analyze several projects that were successful, and non-successful delivered, in order to create a project success profile (PSP), and a project failure profile (PFP), and look for RBPs that are associated to each of the project profiles. | Method: Measure the deviation between the evolution of the ongoing project-phase, and the critical success factor for the respective project-phase. |
| Requirements:<br>Project data-related, from projects that were successful, and unsuccessful delivered, must be available for each project phase according to Table 3.<br>The number of projects that were successfully delivered, must not exactly be the same, as the number of projects that were unsuccessfully delivered.<br>The Model previews, a *n* number (finite undefined number) of different organizations that can be analyzed. One possible arrangement can be:<br>The organization that delivers a project (service provider) and, the organization that ordered a project (customer). They can still be named simply as Team A, and Team B respectively.<br>The criterion that defines a successful, and a failure delivered project is not to be defined by the proposed model, rather given by the PMI (PMI, 2017).<br>All projects need to have the same number of phases throughout their lifecycles for a given analysis. | Requirements:<br>Critical success factors must have been identified at Part 1 of the model.<br>The Model previews, a *n* number (finite undefined number) of different organizations that can be analyzed. One possible arrangement can be:<br>The organization that delivers a project (service provider) and, the organization that ordered a project (customer). They can still be named simply as Team A, and Team B respectively.<br>All information regarding the way project people dynamically interacted, for the respective project-phase where an outcome estimation likelihood is to be calculated, needs to be available according to Table 3. |

### 3.7. POL Model Requirements

So that the model can function, a set of requirements is needed for Part 1 and Part 2. These requirements are illustrated in Table 2.

The necessary information regarding the dynamic interaction of project people will be collected through three different social interaction tools (SIT). This necessary information is illustrated in Table 3. Meetings refers to F2F (Face to Face), or Web project meetings that occur in each phase of a project lifecycle. Mails refer to all the project related information emails that were exchanged between the project people, in each phase of a project lifecycle. Questionnaires, will only be used in the phases of the project lifecycle, were a third organization is outsourced to accomplish project activities. Usually, Questionnaires are applied at the execution phase of a project lifecycle. The data collected through these three SITs will then be analyzed, using a set of tools and techniques based on social network analyses, and statistics that enable us to measure the dynamic interactions of project people across all phases of a project lifecycle.

**Table 3.** Necessary information regarding each of the social interaction tools.

| SIT | Necessary Information (Projects Logs) |
|---|---|
| Meetings | - Total number of project meetings in each project phase, across a project's lifecycle<br>Total number of participants in each project meeting, in each project phase, across a project´s lifecycle<br>Name, Project Official Role, and belonging Team from each of the participants |
| Mails | ○ Total number of emails sent/received in each phase of a project, across a project´s lifecycle that relate to project information matters<br>Name, Project Official Role, and belonging Team from each participant that sent / received emails project related information.<br>Categorize emails according*:<br>Mails sent in seeking for help, or advice regarding project information related matter<br>Mails sent, providing help, or advice, regarding project information related matter<br>*Mails need to be accessed and identified either by their subject or content, as being seeking or providing help type. |
| Questionnaires | ○ To be applied, case a third Team (possibly called as Team C) takes part at one, or more than one phase, of a project lifecycle, and usually works on behalf (outsourced) of one of the main Teams, A and B.<br>Conduct a SNA by applying following two questions to the third Team members:<br>Question 1: If you have a problem, or question regarding (x)* that is important to execute your project activity, who do you usually go for help, among the project people of Team A and Team B?<br>Question 2: If you are just about to start the execution of a project activity or task, but you want to make a double check before you go ahead, or even present what you consider to better solution, whom do you turn to, to get approval and final decision among the project people of Team A and Team B?<br>*(x) is a project related activity or task, to be named as the project execution phase occurs. |

The quantitative results outputted by the SNA-metrics and statistics will be used to characterize five global collaboration types (5-GCT) that usually occur between project people, across all project phases of any project lifecycle, for both PSP and PFP. They are, (a) communication and insight, (b) internal and cross-collaboration, (c) know-how and power sharing, (d) clustering (variability effect), and (e) team efficiency. In Table 1 is illustrated a detailed description on the five global collaboration types (5-GCT) that usually occur between project people, across all project phases of any project lifecycle.

In Figure 2 is illustrated the overall process of the POL model for Part 1. The proposed model is not constrained to a determined number of phases of a project lifecycle. However, four phases will be adopted, which represent the generic phases of a project lifecycle recommended by the PMI (PMI, 2017). First, the required information for each project phase is collected (according to Table 3) for both project outcome types (success and failure). Then, the collected information regarding both project outcome types (success and failure) go through process of analysis, by the application of social network analysis tools and techniques and statistics, which allows the project people's relational data (dynamic interactions) to be measured for both project outcome types. After that, a project success profile and a project failure profile are created. Then, the results obtained for both PSP and PFP are used to characterize the five global collaboration types (5-GCT). After that, if different RBPs (regarding each of the five global collaboration types) are to be found for each project outcome type (success or failure), then the critical success factors for the project-phase that is being analyzed have been identified! For Part 2 of the proposed model, the process is quite similar as for Part 1, but some steps are suppressed. As previously said, Part 2 of the model only makes sense if both Part 1 has already been run, and critical success factors have been identified for a certain project phase; otherwise, there will be no data to compare with. For example: Part 1 has been run, and project-phase critical success factors for all project phases have been identified. A new project is ongoing and finds itself at a certain point in time (AP) within the Phase 2. In this case, the model should be run with the requirements (by the pol

model) and availability (since Phase 2 of the new project started until *AP*) for Phase 2, and the results should be compared with the critical success factors identified for Phase 2. After the comparison has been made, decisions can be made as to whether there is a need for implementing corrective actions or not.

### 3.8. POL Model Implementation

As previously said, the number of phases adopted in this work will be four. They are, *starting the project*, *organizing and preparing*, *carrying out the work,* and *completing the project.* These phases are adopted from the recommendations of the PMI. For the demonstration of the implementation of the POL Model, a real application case of was used, and it will be illustrated in the following pages. For this illustrative case, a generic project with four different phases has been chosen, where Teams A and B are expected to collaborate to ideally successfully deliver the project. In each project phase, meetings, email-exchange communication, and questionnaire data (in this case only at Phase 3) were recorded according to Table 3. The implementation goes as follows:

### 3.8.1. Defining the Official Project Formal Structure

First, the model previews the establishment of a formal organizational chart at the very beginning of the first phase of a project lifecycle (Figure 5), or even before the mentioned phase, and is to be set one time only.

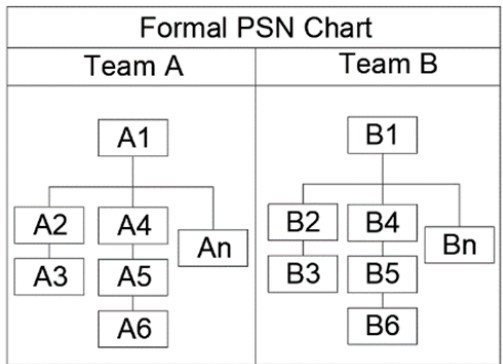

**Figure 5.** Organizational formal chart—model implementation.

In Figure 5 is the formal organizational chart of two Teams—Team A and Team B—that are collaborating across all the different phases of a project lifecycle in order to deliver a project. The formal importance degrees—known as the official line of command of both Teams A, and B—are represented on the formal organizational chart at Figure 5 which is top-down (from A1 down to A6, as an example of Figure 5) ranked; on the top are the most important project roles, and at the bottom are the minor project roles. The formal chart displayed in Figure 5 is where who plays what role is to be seen, and who does what, throughout the project lifecycle. Furthermore, for the POL model is the setting of the base-team that is expected to deliver the project throughout all the phases of the project lifecycle.

### 3.8.2. POL Model Implementation of Phase 1—Starting the Project

In Figure 6 is illustrated the implementation process for project Phase 1—*starting the project.* In this phase (as well for the following project phases), there must be two well-defined points in time, representing where the phase starts and where the phase ends. Those are represented by the points at red *START* and *END* at Figure 6. Across this phase, a set of project meetings did occur, and the information regarding those meetings is to be collected according to Table 3. Project meetings are named events (*E*) and are coded as: *E1_1*, where for example: E_1 = event of Phase 1, and _1 = event, or meeting number one. There may exist E1_*t* events in this phase, as for all the other project phases.

For example, in Event 1 (E1_1) at Figure 6, the project people who participated were A1, A2, and A5, from Team A, and B3, B4, B5, and B6 from Team B. The same interpretation is to be made for the remaining Events E2, E3, E4, and E5 (E1_*t*). Figure 6, under the project lifecycle curve, contains a table representing the PSNVar_1, which a list of the actual participants, who left, who came anew, and who re-came, at any given project meeting of project Phase 1. As an example, let us consider Meeting E1_2, as a present (actual) meeting for an instant. At Meeting E1_2 (the second project meeting form Phase 1 of the project illustrated in Figure 6), project people A1 and A3 from Team A, and B1 and B2 from Team B, were present. This information is displayed in the box above the red *START* point. In the table where—Teams A, B, IN, OUT—are displayed (under the project lifecycle curve), for Meeting E1_2, project person A3 from Team A is categorized as IN. This means that project person A3 is present at Meeting E1_2 (present meeting) and was not present at the previous meeting (E1_1). The categorization IN, means that project person A3 just came into this meeting (E1_2), and at the same time did not attend the previous meeting (or previous meetings). As it can be seen, project person A3, was not present at Meeting E1_1. Still, at Meeting E1_2 (the present meeting for this case), project people A2 and A5, are categorized as OUT. This means that they were at the previous meeting (E1_1) but not at the actual meeting (E1_2). The same interpretation is to be made for project people of Team B. Let us now consider Meeting E1_3 as a present meeting. At this meeting project person A1 from Team A is the only element present at the actual meeting. From Team B, project people B1 and B4 are present. At this present meeting (E1_3), project person B4 is categorized as IN, which means that B4 was not present at the previous project Meeting E1_2. Project person A3 is now categorized as OUT, meaning that B2 was present at the previous project meeting (E1_2), but is not present at the actual Meeting E1_3. The same goes for project people B2 of Team B. Still, at the present meeting (E1_3), no project people of Team A are categorized as IN. This means that at the present meeting (E1_3), there are no project people from Team A that simultaneously did not participate on the previous meeting and participate at the actual meeting. In other words, this means that there are no *newly* arrived project people from Team A, to the present meeting since the last meeting. Considering now, Meeting E1_4, as the present meeting, there are no project people categorized as OUT from either Team A or B. This means that, from the previous meeting, up to the present meeting, no project people left the meetings circuit from either Teams. In other words, this means that the project people that participate at Meeting E1_3, also are participating at Meeting E1_4. Considering now Meeting E1_*t* as the present meeting (which is as well the last meeting from the Phase 1 of the project lifecycle), project people A1 and A2 from Team A, and B1 and B4 from Team B participated at the actual event. In this case, for Team A, there is no project person categorized as IN or OUT. This means that the project people that participate at the actual meeting (last meeting of Phase 1) are the same project people that participated at the previous project Meeting E1_4. In such cases, the variability from one meeting to another meeting is zero, regarding the project people attendance.

For a given actual meeting, project people categorized as IN, are considered *new* at that given meeting. This *new* categorization has two different interpretations and is related with the restart (**R**) step, illustrated in Figure 7. The first interpretation of *new*, is related to a project person, that is taking part for the very first time at any given project meeting. The second interpretation is related to a project person, that is taking part a given project meeting, but it is not the first time that this project person participates in meeting of the given project. In other words, that project person had already participated in some previous project meetings of the given project. This concept is explained in detail in the dynamic variability cycle illustrated in Figure 7.

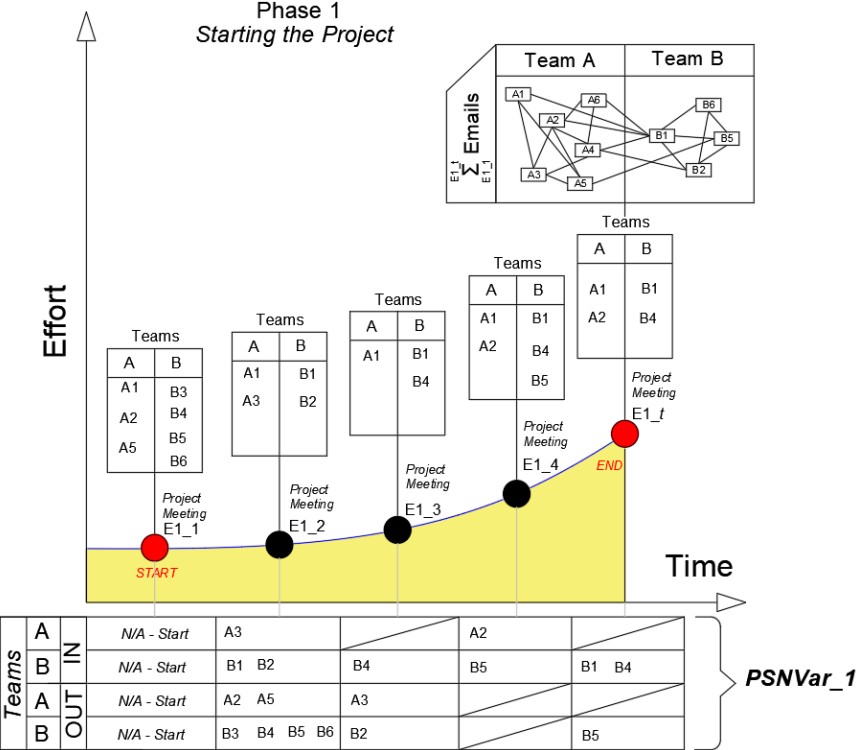

**Figure 6.** POL model implementation for Phase 1, for both success and failure project outcomes. Example with five meetings.

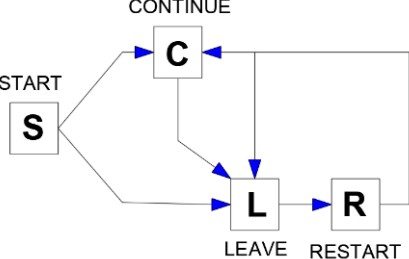

**Figure 7.** SCRL (Start, Continue, Restart, and Leave) dynamic variability cycle meetings' participation assiduity.

The PSNVar (Project Social Network Variability) is based on the project people variability cycle illustrated in Figure 7 and aims to understand to which extent the project social network variability impacts a project outcome. In other words, the PSNVar will quantitatively capture the variability of project people regarding the meetings attendance cycle (Figure 7), which consists of project people starting participation in a meeting (S), leaving (L), or continuing (C) with the successor meeting, and the cycle closes in the case of restarting (R) the meeting attendance after having left the previous meeting. Still, in Phase 1 of Figure 6, the POL model previews the collection of all project-related information emailed between the participating Teams (in this case Team A and B) according to Table 3. This is illustrated in the box above at the upper right corner of Figure 6.

### 3.8.3. POL Model Implementation of Phase 2—Organizing and Planning

In Figure 8 is illustrated the implementation process for project Phase 2—organizing and preparing. The process is the same as for Phase 1, excluding the part of defining the formal project chart, which has already been defined at Phase 1.

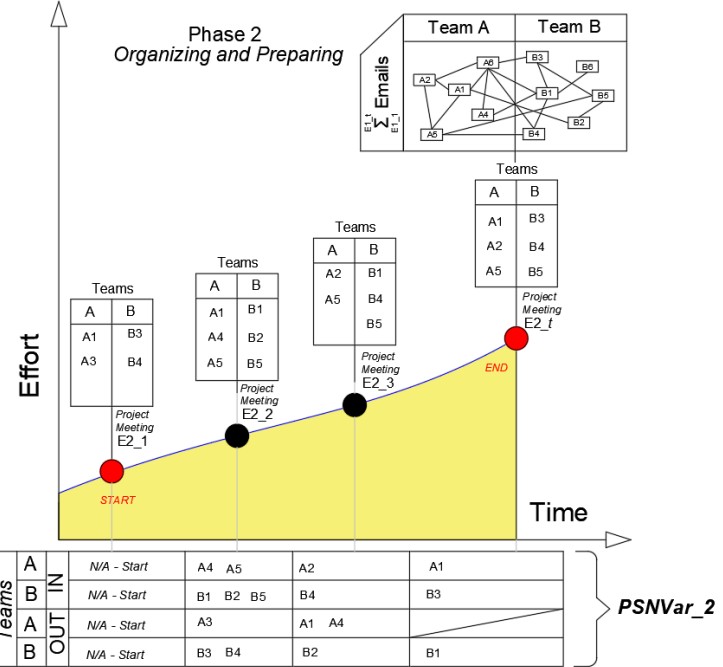

**Figure 8.** POL model implementation for Phase 2, for both success and failure project outcomes. Example with four meetings.

### 3.8.4. POL Model Implementation of Phase 3—Carrying out the Work

In Figure 9 is illustrated the implementation process for project Phase 3—*carrying out the work.*

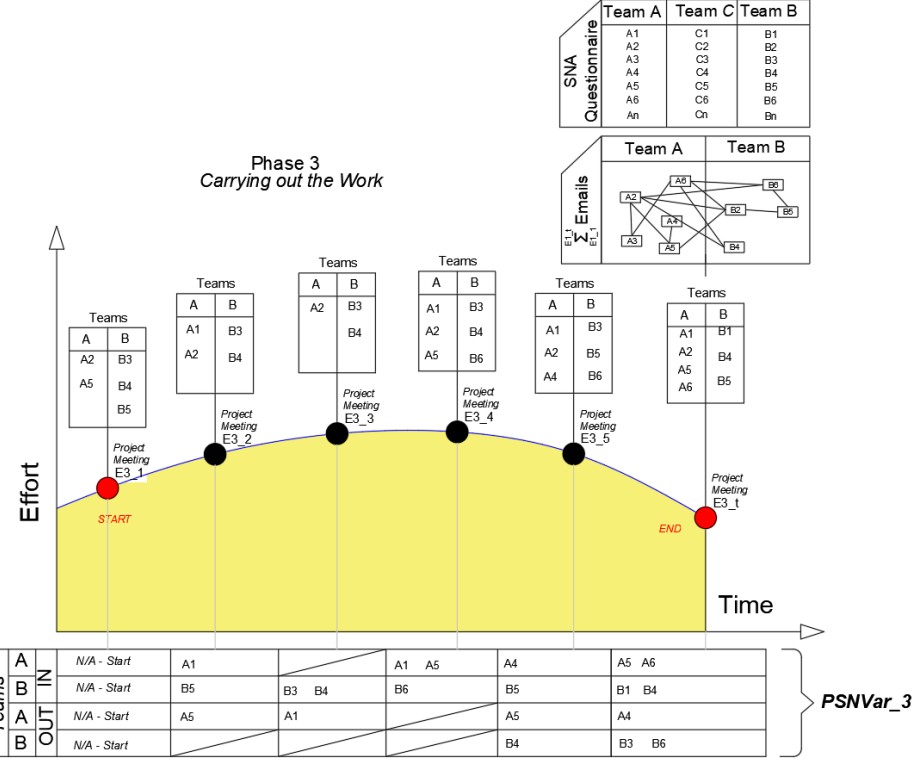

**Figure 9.** Model implementation for Phase 3, for both success and failure project outcomes. Example with six meetings.

The process for Phase 3 illustrated in Figure 9 is exactly the same as for Phase 2 illustrated in Figure 8, except that in this phase a new team (Team C)—which is previewed by the model at Figure 2—is taking part in the *carrying out the Work* project phase. The data to be collected regarding the third-party team are to be according to Table 3 at the questionnaires line.

3.8.5. POL Model Implementation of Phase 4—Completing the Project

In Figure 10 is illustrated the implementation process for project phase 4—*completing the project*. The process is the same as for Phase 2.

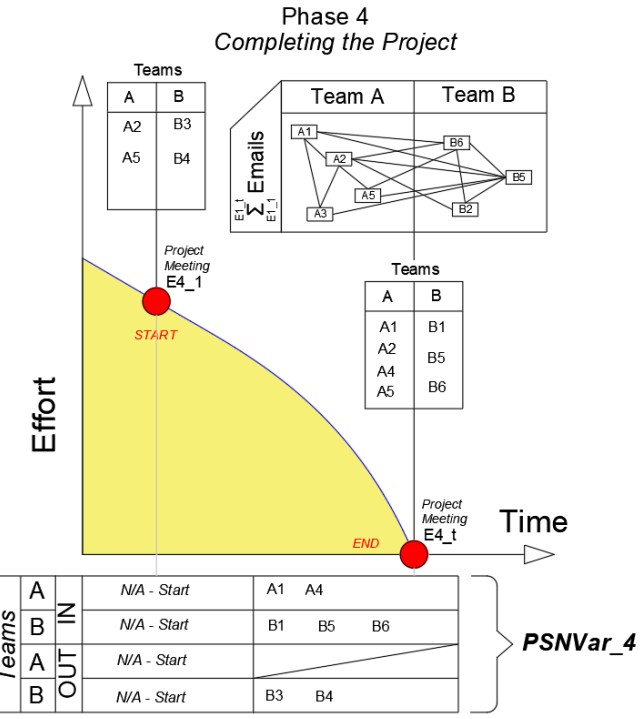

**Figure 10.** Model implementation for Phase 4, for both success and failure project outcome types. Example with two meetings.

*3.9. Model Ethical and Legal Considerations*

The proposed model accesses and analysis project related information that flows across the different project teams across a project lifecycle that may be considered confidential, and not desired to be accessed, and/or exposed. Therefore, the implementation of the model in its total plenitude (as it is designed) is totally dependent on the acceptance of competent authorities, at the organizational and ultimately nation level that administer the legal and ethical respective issues. However, all the project people that participate in a project that is to be monitored by the proposed model, should be aware of it, before the project starts.

**4. Proposed Model Metrics**

The proposed model previews the application of seven centrality-based metrics. They are presented at Table 4. Although it was not the objective of this work to extensively detail the metrics applied in the model, a brief description is presented at Table 4. At Table 4, for each metric the source of information (SITs) is indicated, as well as the respective global collaboration type associated. For example, the global collaboration type communication and insight is characterized and measured for both projects, successfully and unsuccessfully delivered, by analyzing meetings and emails exchanged with project-related information. For this collaboration type—communication and insight—the model previews the application of three metrics. They are: role attendee degree (which will analyze project

meetings information), internal email cohesion degree (which will analyze exchanged information), and feedback degree (which will analyze information email exchanged). The model previews individual and collective analysis types. Individual analysis is when either certain official project roles, such as, project managers, engineers, experts, or other (customizable), are isolated analyzed in the project social network. Collective means that a group (team) or all the project people will be analyzed.

**Table 4.** POL model proposed metrics.

| Metrics | Brief Description | 5-GCT |
|---|---|---|
| | Individual Analysis Type—*Meetings* | |
| 1-Role attendee Degree | Analyze the project meetings-participation rate, from Official Project Roles of the Teams that take part in the accomplishment of a given project. The participation rate will be outputted by an average variation (linear regression) across a project phase, previewing four different results. <br><br> Objective: To find out if project success outcome is somehow correlated with: <br><br> High participation rate of Official Project Roles in project-meetings, at the first half (beginning) of a given project phase? <br> High participation rate of Official Project Roles in project-meetings, at the second half (at the ending) of a given project phase? <br> Constant participation rate of Official Project Roles in project-meetings, at the first half of a given project phase? <br><br> If a)—evolution is characterized by a negative (-) evolution across time, within a given project-phase <br> If b)—evolution is characterized by a positive (+) evolution across time, within a given project-phase <br> If c)—evolution is characterized by a constant (0) evolution across time, within a given project-phase <br><br> Case 1 c)—Constant Full (when the Official Project Roles participated in all project meetings <br> Case 2 c)—Neutral Constant, (when the Official Project Roles did not participate in all project meetings | |
| 2- Mail cohesion Degree | Individual Analysis Type—*Meetings* <br><br> Analyze the Centrality Structural degree from Official Project Roles from the participating Project Teams, regarding the email communication network. <br><br> Objective: To find out if project success outcome is somehow correlated with: <br><br> High, or Very High Centrality in the email communication network of Official Project Roles. <br> Low, or Very Low Centrality in the email communication network of Official Project Roles. <br><br> This is outputted by the SNA In-degree metric according to [43]. The result for this metric is a numerical value (index), varying from 0 (minimum), to 100 % (maximum) for the two cases. <br><br> Collective Analysis Type—*Mails* <br><br> Analyze the Cohesion degree from all Official Project Roles from the participating Project Teams, regarding the email communication network. <br><br> Objective: To find out if project success outcome is somehow correlated with: <br><br> High, or Low Centrality in the email communication network of All Official Project Roles. <br><br> This is outputted by the SNA density (1) metric [43]. The result for this metric is an absolute numerical value, varying from 0 (minimum), to 100 % (maximum). | Communication and Insight |
| 3-Feedback Degree | Collective Analysis Type—*Mails* <br><br> Analyze the Reciprocity degree (Feedback) between the different participating Project Teams, regarding the email communication network. <br><br> Objective: To find out if project success outcome is somehow correlated with: <br><br> High, or Low Email project-related information Feedback between different project Teams regarding the email communication network (All Official Project Roles) <br><br> This is outputted by the SNA reciprocity metric [43], which is based on the difference from Out against the In-degree [43]. The result for this metric is an absolute numerical value, varying from 0 (minimum), to 100% (maximum). | |
| 4-Information Seeking / Providing Degree | Collective Analysis Type—*Mails* <br><br> Analyze the seeking / providing, of project information-related between the Teams that take part in a project accomplishment. This is outputted by the SNA in-degree metric, respectively seeking and providing. <br><br> Objective: To find out if project success outcome is somehow correlated with: <br><br> High or Low Project-related Information seeking degree? <br> High or Low Project-related Information providing degree? <br><br> This is outputted by the SNA In-degree, and Out-degree metrics [43]. The result for this metric is an absolute numerical value, varying from 0 (minimum), to 100% (maximum). | Internal and Cross-Collaboration |

**Table 4.** *Cont.*

| Metrics | Brief Description | 5-GCT |
|---|---|---|
| | Collective Analysis Type—*Questionnaire* | |
| 5-Action Key Players | Analyze the Centrality Structural degree from the Teams that take part in a project accomplishment, from an external point of view, regarding know-how sharing and power over the informal network. <br><br> Objective: To find out if project success outcome is somehow correlated with: <br><br> High or Low Centrality regarding power and know-how share of involved teams at the accomplishment of a project <br><br> This is outputted by the SNA in-degree metric [43]. The result for this metric is the project team that is more, *voted*, by the external point of view | Know-how sharing and Power |
| | Collective Analysis Type—*Meetings* | |
| 6-Meetings Cohesion Degree | Analyze the variability evolution within a project phase, regarding the participation rate of all Official Project Roles (project team set) in project meetings, for each of the project teams that take part in a project accomplishment. This is outputted by the SNA *PSNVar* an original developed metric, based on weighted average-degree [43]. <br><br> Objective: To find out if project success outcome is somehow correlated with: <br><br> **Constant** (=0) no change on the project set team across a project phase <br> **Non-Constant Positive** (= +) tendentially, are entering new project people across a project phase <br> **Non-Constant Negative** (= -) tendentially, are leaving project people across a project phase. <br><br> The output for this metric is an evolution across time (within a project phase time period), calculated by a simple linear regression from all the individual results of the metric *PSNVar* for each project meeting within a project phase. The formula is illustrated as follows: $$V_{(Et)} = \frac{WL_{(Et)}}{TPP_{(Et)} \times Et}$$ Where: <br> V = Variability of the Project Social Network <br> Et = Event number (Project Meeting), where Et = 1,2,3, … ,TE <br> TE = Total number of events (Project Meetings) occurred within a project phase <br> TPP = Total number of project people that participated in an Event Et. <br> WL = Total commulative value of weighed links, from each project people´s total degree in each Event Et. For example, if in an Event X, project people 1, and 2 participated in, the link between them is of value 1. If at Event X + 1, project people 1, and 2 participated in, the link between them is of value 2. | Clustering (variability effect—PSNVar) |
| | Collective Analysis Type—*Mails* | |
| 7- Transferring Speed | Analyze, in average, the project information-related transferring speed, when requested all Official Project Roles in all project related email exchange. <br><br> Objective: To find out if project success outcome is somehow correlated with: <br><br> Feedback Speed when answering a question or providing project information-related through the email network communication <br><br> The output for this metric is an average value displayed in hours, ranging from "100%" (meaning an instantaneous answer reply has been made < 1 h) up to a maximum of the project time duration "0%," for cases where feedback is not finding during the lifetime of a project. The metric to be used in this case is the Out and In-degree [43], for each single pair (question vs answer) time-attached. | *Teamwork efficiency* |

## 5. Benefits and Limitations of the Proposed Model

The proposed model allows organizations to learn from past experiences—lessons learned—uncovering, understanding, and measuring the reason(s) that led to failure (regarding the way different people from different organizations interacted, as they performed in networks of collaboration across a project lifecycle) to avoid their repetition, and replicate behavioral patterns associated with a project's successful outcome (critical success factors) in upcoming projects. Being able to measure (quantitatively), how those interactions contribute to a failure or to a successful project outcome, enables organizations to design strategies that are more driven by data and insights, than the traditional approaches, such as gute feeling, and biased influenced advices or hunches, when it comes to changing the way organizations work, for example, by addressing leadership behaviors, and diversity and inclusion issues. This in turn, will enable organizations to make decisions more accurately, which directly positively contributes to the economic, environmental, and social sustainability inasmuch as, once known, the reasons for project success and failure can be used to replicate the success and avoid or eliminate failure, thereby avoiding unnecessary risks, and saving resources. This drives organizations to become leaner oriented. The implementation of an automated collecting and processing data system is a huge step forward in the digital transformation process, regarding the logging of project people's dynamic interactions in networks of collaboration across a project lifecycle, by efficiently using available

technology. By uncovering the different dynamic behavioral patterns that occur across the different phases of a project lifecycle, such as employees of the organizations involved delivering projects, the proposed model allows one to trace a working culture profile that mirrors how organizations work in collaborative networks that are associated with success. This is achieved by analyzing data arriving from communication/collaboration channels—project meetings, emails, and questionnaires—that organizations use to deliver projects. Collecting data from project meetings and emails is considered a non-invasive method and presents two major advantages against the questionnaires collection data type. First, the data is almost totally free from bias, or less biased influenced. Second, no employee down-time is expected to occur, because as soon as the model is implemented the collecting process is started automatically. These two factors represent two big advantages. First, the collected data mirrors more, the reality of the collaboration between the participating organizations, and second, makes the model more economically viable. The identification of critical success factors is a crystal-clear process. This means that critical success factors are identified by opposition to critical failure factors, because they must be a unique, repeatable behavior pattern. Still, critical success factors are quantitatively measured, and not only qualitatively measured, where the results/conclusions are not a direct function of the size of a given network of collaboration, but rather the interactions between the participating organizations. This allows, in a very clear way, one to observe and interpret the different dynamic behavioral patterns that occur across the different phases of a project lifecycle, which greatly facilitates intervention in order to take corrective actions. Once the model is in full operation, it can be seen as a self-learning system; it refines and monitors best practices, regarding the repeatable behavioral patterns that are associated with success, as people work in networks of collaboration towards the accomplishment of a common objective. This enables an organization to be more responsive to changes, at the very early stages of the development of a product solution or project phase. The implementation of the proposed model in an organization introduces to a certain extent, a sense of awareness among the employees, which immediately triggers in them, willingness for engagement towards a more accurate way of doing things. This factor highly contributes to cementing the fact that flexibility and adaptability are vital to survival, and to improving the way organizations work. Furthermore, by the application across a substantial time period of the proposed model, it will be possible to establish the real importance of informal networks of collaboration by comparing the informal against the formal (formal chart) networks of collaboration regarding project success or failure outcomes. Finally, the model can be applied to any project type if it respects the model requirements and structure presented throughout this work. Although the advantages by far outweigh the disadvantages of the proposed model, when tracing a PSP and a PFP, by applying mathematical operations, one will naturally be influenced by the nature that comprises the mathematical operations. The initial phase of the implementation of the model may be slow—namely, regarding the implementation of a data-collecting culture, filtering project information-related emails, and clearly identifying and defining transitional phases between different project phases—as organizations collaborate across a project lifecycle. Data collected through questionnaires, pose some natural issues; namely, regarding how trustable the data is. In fact, this data accuracy dependents almost entirely on respondents' good-will, to provide honest and non-biased information. Project organization dynamic interaction chain-break represents another limitation of the introduced model. This means that, it often happens that project related information is discussed via other type of communication/collaboration channel, such as through phone calls, or thought informal corridor meetings. These types of communication/collaboration channels are not covered by the present version of the proposed model. Finally, because the model previews the assessment of what many may consider confidential project information, its implementation will be always conditioned by legal and ethical aspects both at organizational, and country level.

## 6. Conclusions and Further Developments

The proposed model in this work, contributes to the organizational Transformation scientific field, namely, to the project management field, regarding the project people risk management strategy.

It introduces a new approach concerning how to identify and measure the impact of project people dynamic interactions across a project lifecycle, regarding its influence in a project´s outcome (failure, or success). The model was developed base on three scientific fields (project management, risk management, and social network analysis), with special focus on the application of existing, and new social network analysis Centrality Metrics Graph theory-based, which according to latest research described at sub-chapter 2.2.3, are the ones that properly / better uncover and measure the importance of project people having a more, or a less central location within a project social network, which in turn is often associated with influence, prestige, control and prominence, coordination and decision-making, in project environments. Network tie strength, or familiarity—a direct consequence of being central within a project social network—according to latest research is also an indicator of the importance of a project people within a project social network. This is also captured, with the adapted-develop Clustering (variability effect—PSNVar) metric, illustrated in Table 4. The proposed model is divided in two parts. At part 1, the proposed model, will identify and quantitatively measure, critical success factors from past delivered projects regarding to how project people dynamic interacted across a project lifecycle. At part 2, the model, will use identified critical success factors, to provide guidance to upcoming projects, in order to enhance the chances of project success outcome. The approach of the proposed model is aligned to what renowned people and institutes argue [33,34], which is, more research and investigation should be directed to understand how people behaviors and dynamic collaboration patterns may influence outcomes. The propose model, to a certain extent is a tool that enables the management of the project ambiguity risk type, which is characterized essentially by a lack of knowledge or understanding and can be fought by essentially learning from past experiences—lessons learned. In fact, the proposed model, essentially tackles one of the most important factors that enables organizations to innovate, improve, optimize, and gain competitive advantages—learning from past experiences, namely, mistakes (lessons learned)—so that organizations can eliminate or avoid traps/failure behaviors, and replicate (and improve) successful behavioral patterns in upcoming projects. The proposed model focuses its attention on what is today considered one of the fundamental pillars of performance and innovation—social capital—by analyzing the social relationships between project people that ultimately enables value creation, as is proposed by several authors [92,93], simultaneously contributing with valuable insight to the people data management field, usually known as people analytics or people big-data. The proposed model strongly contributes to the achievement of a competitive advantage (to a certain extent acting of the *differentiating side* according to the popular Porter's model [94]) of an organization in a sustainable way, essentially because by quantitatively identifying project critical success (and failure by direct opposition) factors (Part 1 of the model) regarding the dynamic interactions of project people across the different phases of a project lifecycle, instead of the traditional static approach of analyzing project people's performance rates, essentially based on project risk registers and lessons learned files from delivered projects, will enable organizations to take a *real-time*, necessary adjustment measures, by continuously quantitatively comparing a real ongoing project evolution, against a desired (based on identified CSFs) project evolution (part 2 of the model) in each phase of a project lifecycle. This in turn, will enable organizations to faster adapt and respond to changes by taking a more *data-driven decisions* approach, which will ultimately strongly contribute to more rigorous and accurate management of intangible and tangible project resources, by exactly knowing how much and where action should be taken. However, continuous research should be done regarding the development of new or improved existing metrics, based on social network analysis theory.

**Author Contributions:** Author M.N. carried out the investigation methodology, writing—original draft preparation, conceptualization, the formal analysis, and collected resources. Author A.A. contributed with the conceptualization, writing—review and editing, the supervision, and final validation. All authors have read and agreed to the published version of the manuscript.

**Funding:** This research received no external funding.

**Conflicts of Interest:** The authors declare no conflict of interest.

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
