# Peer review of "Applying Social Network Analysis to Identify Project Critical Success Factors"

_sustainability, doi:10.3390/su12041503_

Round 1

Reviewer 1 Report

Dear Authors:

The work presented is very interesting from the perspective of human resources management in projects. It correctly analyzes the elements that can influence success and poses a correct review of the state of the art.

The model he proposes is based primarily on parameters related to critical success factors and communication. It mainly analyzes three social interaction tools; meetings, emails and questionnaires, but it focuses on the first two.

It also proposes a definition of the organizational structure for the project, which is reflected in the PMI standard Defining the Official Project Formal Structure: In PMI it's showed in Project chart in kickoff meeting (PMBOK), so I don't find originality in this pulled apart.

About regarding the Benefits & Limitations of the proposed model:

The model is based on formal communications by mail, this can interfere with the confidentiality of the information. These issues must be incorporated into the contract and cannot be left to the discretion of the person. It is also true that informal communications (impromptu meetings or coffee talks) must be documented. All information must be treated properly to generate the lessons learned from the project.

It is recommended that you indicate aspects related to project risks and phases, an issue that is relevant in project management.

The conclusions presented in the work should be reviewed, since there is no direct relationship between those shown and what is presented in the different sections.

I agree that it is necessary to provide metrics that help quantify the effect of the use of collaboration networks, both formal and informal, so I invite you to do so.

It is recommended that you analyze the aspects of PMBOK Section 10: Project Communications Management and review the processes in the life cycle.

Kindly regards.

Reviewer 2 Report

My comments are based on few issues: conceptual; representation; editing/writing errors.

Firstly, on page 3 is mentioned a paper from 2019 (line 138) related to the Project Outcome Likelihood Model (POL). In order to make a good delimitation (if any) of this model proposed in 2019 (same authors) and the present model (described in the paper) it is necessary and important to show in detail what the 2019 proposal contains in relation to the current proposal. Are there any important parts taken from the previous approach? Basically, they have the same name. Is important to know the relation between these two proposals (2019 vs.2020).

Also in the paragraph 3.2/ page 9 (I will come back later with a comment related to the editing errors) in describing details for the Model function principles, are mentioned two projects A and B. In this respect, is important to include some info about these two projects. There are similar projects in terms of objectives, resources, outcomes, duration etc.? These two projects were for real? In the same time are mentioned two organizations (two teams) involved in the phases of each project. What happens with the approach when we have three ore more than three different organizations (teams) that are participating in fulfilling the tasks and/or project activities?? We could extend the findings of this model also to such particular cases?

On the same thread of thinking logic, if we have big differences in terms of project people number in each of the four phases mentioned above, is the density formula (1) from page 10 sufficient to support the conclusions? For instance, we could have for project X, in the second phase, a group of 15 project people involved (with 3 or 4 organizations/ teams) and for project Y, in the second phase, a group of 5 project people (with two organizations/teams)! Could be sustained similar findings? Could comparisons between projects X and Y be maintained in the same logic of the approach?

The calculation for the application of formula (1) for phase 3 is missing in both cases (project A and project B). Is only mentioned on line 482 the value for the density in the case of project A.

A comment now regarding the way of notation. It can be found at 3.2.4,3.4, 3.7.1 or 3.7.2 notations as: team A and team B. But also there are mentioned projects such as: Project A and Project B (3.2 at page 9). Even there are different things when there a mix of these issues there is a possibility of confusing when the readers are not so focus to follow very carefully all descriptions. It is advisable to revise these aspects in order to avoid any misunderstanding.

At a certain time, on page 11, is mentioned a need for another SNA metric because the density metric is insufficient and unrepresentative to be used. In this respect, shouldn't the number of emails and meetings made (i.e the volume of interactions in a given period of time) be taken into account?

Even if it is not directly related to the paper subject, a legitimate question may arise: What is the ratio or the percentage regarding the influence on the project success between project formal networks and project informal networks?

All these issues requires at least few ideas or comments in the area: Conclusions & further Developments. Also the part called Benefits & Limitations of the proposed models needs to be improved!

Finally, please pay great attention to the numbering of the figures. Figure 1 appears two times, figure 3 also appears two times and figure 4 appears two times as well. After the figure 5 is jumps directly to figure 9 etc. Figure 2 from page 18 is in fact figure 5 and so on.....

Please pay also attention to the spelling because there are few small errors. For instance, in lines 403 and 486 you have Costumer instead of Customer. In line 696 you have precious instead of previous. Also for the punctuation marks there are some errors. For instance in the line 139 you have semicolon instead of two points.......

There are errors at numbering paragraphs. You have twice 3.2 (at page 7 and at page 9). Therefore, all paragraphs numbers from chapter 3 Model Development and Implementation has to be changed, accordingly.

Round 2

Reviewer 1 Report

Dear Authors:
Thank you very much to consider my recommendations. This paper is better than previous version, I agree with yours new contributions.
Best Regards

Reviewer 2 Report

Basically, I appreciate the job done by the authors in order to make appropriate changes associated with my comments and recommendations from the first review. 

These changes are welcome, but its greatly expanded the paper. I think in at least 2 places the content must be rethought: the Abstract (has now 274 words/ 1926 characters; this is too much and you have to reduce maybe from the first part, first 10-12 lines and make a summary) and 2.2.1 Project Management Challenges (important issue but too long - 1 page).

On the other hand, it is worth introducing the subtitle: 4 Proposed Model Metrics.

Finally, the last 12-13 lines from the Conclusions are dedicated to the relationship POL model - sustainable competitive advantage. The explanations offered do not help much the interested reader to understand how to implement the model in a useful way. This part could be rewrited in a more practical way by explaining concretely how this model can ensure the competitive advantage of a company in a sustainable way.
